# Estrogen-Receptor Loss and *ESR1* Mutation in Estrogen-Receptor-Positive Metastatic Breast Cancer and the Effect on Overall Survival

**DOI:** 10.3390/cancers16173025

**Published:** 2024-08-30

**Authors:** Pieter J. Westenend, Claudia J. C. Meurs, Bertie de Leeuw, Robert C. Akkers

**Affiliations:** 1Laboratory of Pathology, 3318 AL Dordrecht, The Netherlands; bertiedeleeuw@gmail.com (B.d.L.); r.c.akkers@resultlaboratorium.nl (R.C.A.); 2CMAnalyzing, 6904 HC Zevenaar, The Netherlands; c.meurs@cmanalyzing.nl

**Keywords:** breast cancer, metastasis, estrogen-receptor, estrogen-receptor loss, ESR1 mutation, survival

## Abstract

**Simple Summary:**

In metastatic estrogen-receptor (ER)-positive HER2-negative breast cancer, resistance to endocrine therapy can be caused by ER loss and the mutation of *ESR1*, the gene coding for ERs. These mechanisms have been studied in isolation but not in the same population. Here, we studied both mechanisms and their interaction. We found that, in a population of 136 patients, one of these mechanisms was responsible for endocrine resistance in 30% of the patients. *ESR1* mutation was associated with endocrine therapy. ER loss and *ESR1* mutation were mutually exclusive, so testing for *ESR1* mutations can be limited to ER-positive metastases. Furthermore, we demonstrated that ER loss has a negative effect on overall survival, while we did not observe this effect for *ESR1* mutation.

**Abstract:**

In patients with metastatic estrogen-receptor (ER)-positive HER2-negative breast cancer, the loss of ER expression and the mutation of *ESR1*—the gene encoding the ER receptor—are mechanisms for resistance to endocrine therapy. We aimed to determine the frequency of these mechanisms and their interaction. Metastases were retrieved from our pathology files. *ESR1* hotspot mutations resulting in p.(D538G), p.(Y537S), and p.(L536H) were determined by means of pyrosequencing. Clinical data were retrieved from electronic medical records. A total of 136 metastases were available for analysis. ER loss was found in 23 metastases (17%). *ESR1* mutations were found in 18 metastases (13%), including p.(D538G) in 9, p.(Y537S) in 7, and p.(L536H) in 2. *ESR1* mutation and ER loss were mutually exclusive (*p* = 0.042), and *ESR1* mutation was associated with endocrine therapy (*p* = 0.002). *ESR1* mutation was found in two primary breast cancers. *ESR1* mutations are rare in primary breast cancer and develop in metastases during endocrine therapy. Furthermore, ER loss had a statistically significant negative effect on overall survival when compared to patients without ER loss, with a rate ratio of 3.21 (confidence interval 1.95–5.26). No such effect was observed for *ESR1* mutations, with a rate ratio of 1.15 (confidence interval 0.67–1.95). We conclude that ER loss and *ESR1* mutation together account for 30% of the resistance to endocrine therapy.

## 1. Introduction

In patients with metastatic breast cancer, the choice of systemic therapy is largely determined by the hormonal and HER2 receptor status of the primary tumor. Over the past decade, it has been shown that the receptor status of the primary tumor and the metastasis can diverge, known as receptor conversion [1,2,3,4,5]. A wide range of conversion rates has been reported, which may reflect differences in the availability of metastatic biopsies and the duration of follow-up. In a recent systematic review and meta-analysis, the conversion rate for estrogen-receptor alpha (ERα) from positive in the primary tumor to negative in the metastasis was 22.5% (95% confidence interval; 16.4 to 30.0%) [6]. In a retrospective study, the loss of ERα expression in the metastases of ER-positive tumors had a negative effect on overall survival [3]. In a prospective study of the conversion of the receptor status between the primary tumor and metastasis, the treatment plan was altered in 31% of the patients with receptor conversion [4]. Often, the choice of therapy is adjusted after finding a receptor conversion, and this is also recommended in some guidelines [6].

Several mechanisms potentially underlying the loss of ERα expression have been studied. Among these mechanisms are genetic ones involving the *ESR1* gene, epigenetic mechanisms, growth-factor signaling, and post-transcriptional signaling [7]. Alterations at the ESR1 gene level do not seem to play an important role [7]. However, at the molecular level, resistance to endocrine therapy was associated with alterations in genes involved in the mitogen-activated protein kinase (MAPK) pathway, *ERBB2* mutations, *NF-1* loss-of-function mutations, and alterations in genes involved in the regulation of estrogen-receptor transcription [8]. These mechanisms were mutually exclusive with *ESR1* mutation [8]. Mutation in the *ESR1* gene encoding ERα is a well-known mechanism that converts resistance to endocrine therapy in metastatic breast cancer [9,10,11]. Several activating mutations that affect the ligand-binding domain have been identified, and these mutations limit the efficacy of ER antagonists [9,11]. *ESR1* mutations have been found mostly in metastases and rarely in the primary tumor and evolve in response to selection pressure during endocrine therapy. Therefore, it seems reasonable to adjust the therapy according to the detection of the presence or absence of *ESR1* mutations [12,13]. In the AMEERA-3 trial of advanced breast cancer, a selective estrogen-receptor degrader did not show a statistically significant effect on progression-free survival in patients with *ESR1* mutations [14]. Recently, a new selective estrogen-receptor degrader has shown prolonged progression-free survival in patients with metastases harboring *ESR1* mutations [15]. The updated ASCO guideline for ER-positive HER-2-negative metastatic breast cancer recommends testing for *ESR1* mutations [16].

Although several studies have addressed ERα loss and ESR1 mutations, little is known about their interaction. Therefore, our aim was to determine the frequency and association of estrogen-receptor loss and *ESR1* mutation in ER-positive HER2-negative metastatic breast cancer and their effect on time to progression and overall survival.

## 2. Material and Methods

### 2.1. Patients 

Biopsies from patients with distant breast cancer metastases and the corresponding primary breast cancers were selected from our pathology files. Only patients with an estrogen-receptor-positive primary cancer were included. We recorded the age at diagnosis of primary cancer, tumor characteristics, the interval between the diagnosis and the biopsy of the metastasis, the site of the metastasis, the estrogen-receptor expression of the metastasis, and the interval between the date of the biopsy of the metastasis and death or the most recent day alive. Four-micrometer thick formalin-fixed, paraffin-embedded, deparaffinized sections were mounted on Superfrost Plus slides (Thermo Scientific). Sections were stained in an automated stainer (Ventana BenchMark ULTRA) with the ready-to-use SP1 monoclonal antibody (Ventana, Roche, Basel, Switserland) according to the manufacturer’s instructions. The binding of the antibody was visualized with the OptiView DAB IHC Detection Kit (Ventana). Slides were evaluated on a Leica microscope by seven experienced pathologists. Positivity was defined as nuclear staining in 10% or more of the tumor cells, while estrogen-receptor negativity was defined as nuclear staining in less than 10% of the tumor cells. Data on systemic therapy were obtained from patient files. The study was in accordance with Dutch guidelines on the use of residual tissue and patient data.

### 2.2. DNA Isolation and Purification

Formalin-fixed paraffin-embedded (FFPE) tissue was transferred to a lysis solution containing 50 µL of Proteinase K (Sigma Aldrich, Burlington, MA, USA) and 5 µL of 20% Tween80 (SigmaAldrich) in 10 mM Tris-EDTA at pH 8.0 and incubated for 18–24 h in a thermoshaker at 56 °C while mixing at 880 rpm. The samples were incubated at 100 °C for 15 min and then centrifuged at 1400 RPM for 5 min. Next, 200 µL of the sample were transferred to a 2 mL safe-lock tube (Eppendorf, Hamburg, Germany). The DNA was purified using the QIAamp DNA FFPE Tissue kit (Qiagen, Tegelen, The Netherlands) on a QIAcube (Qiagen) following the manufacturer’s instructions. The DNA was eluted in 70 µL of elution buffer, quantified using a BioSpec-nano (Shimadzu, Den Bosch, The Netherlands), and diluted to 2 ng/µL prior to amplification. 

### 2.3. Amplification and Pyrosequencing 

Primers that flank the *ESR1* hotspot region codon 536–538, as previously described by Wang et al. (2016) [17], were used. The primers were 5′-biotinylated-CAAAGGCATGGAGCATCTGTA-3′ and 5′-TGAAGTAGAGCCCGCAGT-3′ and were obtained from Eurogentec, Seraing, Belgium. A 3-step PCR protocol was used to amplify 10 ng of purified DNA (42 cycles: 20 s 95 °C, 30 s 53 °C, and 20 s 72 °C) with the PyroMark PCR Kit (Qiagen). The biotinylated PCR product (10 µL) was bound to Streptavidin Sepharose High-Performance beads (GE Healthcare, Chicago, IL, USA) in a PyroMark binding solution (Qiagen) in a 24-well PCR plate (Qiagen) while mixing at 1400 rpm for 10 min. A sequence primer (5′-CCAGCATCTCCAGCAGCAG, Eurogentec) was designed adjacent to the sequence of the *ESR1* codon 536–538 and diluted in an annealing buffer (Qiagen). Pyrosequencing was performed on a PyroMark workstation following the manufacturer’s handbook (Qiagen).

### 2.4. Statistics

Fisher’s exact test was used to test the association between endocrine therapy, chemotherapy, or the site of metastasis and ER loss or *ESR1* mutation. Time to progression was calculated from the date of the biopsy of the primary tumor to the date of the metastasis biopsy. Overall survival was calculated from the date of the biopsy of the metastasis, and Kaplan–Meier curves were constructed. Differences in survival were tested with the log-rank test. Rate ratios were calculated using the Mantel–Haenszel test and were multivariable using Cox proportional hazards analysis. All analyses were performed in STATA16.1.

## 3. Results

Our files contained a total of 136 metastases of ER-positive HER2-negative breast cancers. The primary breast cancer diagnosis year ranged from 1991 to 2017, while the metastases diagnosis year ranged from 2007 to 2019. The mean age at diagnosis was 58 years (median 57 years; range 33–83 years). The tumors had a mean size of 26 mm (median 20 mm; range 0–110 mm). The majority of the tumors were of no special type (96), grade 2 (67), and presented with axillary nodal metastasis (65). Adjuvant endocrine therapy was given to 89 patients, with 33 receiving tamoxifen monotherapy, 19 receiving aromatase-inhibitor monotherapy, and 32 receiving sequential therapy with tamoxifen and aromatase inhibitor. In five patients, endocrine therapy was not specified. Chemotherapy was administered to 64 patients, with neoadjuvant chemotherapy given to 7 patients. The mean interval between breast cancer diagnosis and metastasis was 84 months (median 62 months; range 0–330 months). The majority of metastases (71) originated from the liver. Table 1 summarizes the additional characteristics.

In 23 metastases (17%), estrogen-receptor expression was lost, and in 18 metastases (13%), *ESR1* mutation was found, including p.(D538G) in 9, p.(Y537S) in 7, and p.(L536H) in 2 metastases. Two *ESR1* mutations, p.(D538G) and p.(L536H), were found in the primary tumors out of 123 tumors evaluated, which were also present in the accompanying metastases. Thirteen primary tumors were unavailable for testing. All metastases with an *ESR1* mutation tested positive for ER. We did not find an *ESR1* mutation in an ER-negative metastasis, which is a statistically significant finding (*p* = 0.042). The occurrence of an *ESR1* mutation was significantly associated with endocrine therapy in 20% of the metastases of patients receiving endocrine therapy and in 0% of those not receiving endocrine therapy (Table 2; *p* = 0.002). There was no association between endocrine therapy and ER loss, which was found in 13% of the metastases of patients receiving endocrine therapy and in 18% of those not receiving endocrine therapy (Table 3; *p* = 0.087). We found no significant effect of chemotherapy on ER loss, which was demonstrated in 19% of the metastases of patients receiving chemotherapy and in 13% of those not receiving chemotherapy (*p* = 0.172). Chemotherapy also did not have an effect on *ESR1* mutation. In 19% of the metastases of patients receiving chemotherapy, an *ESR1* mutation was found, as well as in 11% of those not receiving chemotherapy (*p* = 0.256). Tests for the association between liver metastases and non-liver metastases for ER loss were not significant. ER loss was found in 15% of liver metastases and in 18% of non-liver metastases (*p* = 0.65). Additionally, for *ESR1* mutation, no association was found. The *ESR1* mutation occurred in 17% of liver metastases and in 9% of non-liver metastases (*p* = 0.214).

The results of the time-to-progression (TTP) analyses are shown in Table 4. Three patients were excluded from this analysis because they presented with synchronous metastasis. Age was a significant risk factor, with older patients demonstrating a shorter time to progression in both univariate and multivariate analyses. Tumor size was another significant risk factor; patients with large tumors had a shorter TTP in univariate and multivariate analyses. In patients with ER loss of the metastasis, the TTP was not significantly shorter in univariate analysis, but in a multivariate analysis, taking into account age and tumor size, the TTP was significantly shorter (HR 1.89). Kaplan–Meier curves for TTP according to *ESR1* mutation, ER loss, and age are shown in Figure 1, Figure 2 and Figure 3. The results of the overall survival analyses are shown in Table 5. Thirteen patients were excluded from this analysis, since no information on survival was available. Patients with a metastasis demonstrating ER loss had statistically significantly shorter overall survival compared to those without ER loss (*p* < 0.001). The rate ratio was 3.21 (confidence interval 1.95–5.26). The Kaplan–Meier curves for OS according to age were significantly different, with older patients having a shorter OS. The hazard ratio for ER loss corrected for age was 3.25 (confidence interval 1.92–5.51). For patients with an *ESR1* mutation, no effect on survival was found (*p* = 0.53). The rate ratio was 1.15 (confidence interval 0.68–1.96). Kaplan–Meier curves for OS according to *ESR1* mutation, ER loss, and age are shown in Figure 4, Figure 5 and Figure 6.

## 4. Discussion

The objective of this study was to determine the frequency of loss of ER expression and *ESR1* mutation in metastases from ER-positive HER2-negative breast cancer. Out of 136 metastases, ER loss was found in 23 (17%) and *ESR1* mutation in 18 (13%), 16 of which were de novo mutations. Notably, *ESR1* mutations were exclusively found in ER-positive metastases, which is a statistically significant finding. The acquisition of *ESR1* mutations was found to be associated with having received endocrine therapy, while ER loss did not show a clear association.

The reported 17% loss of ER expression in metastases is consistent with previous findings, although it falls within the lower range [6]. A prior study also noted a site-specific difference in receptor conversion for ER, which was lower than average in the liver [6]. In our series, 71 out of 136 metastases (52%) originated from the liver. However, we did not observe any significant difference between liver and non-liver metastases. The molecular mechanism responsible for ER loss has not been fully elucidated, although several possibilities have been suggested, such as selective pressure from endocrine therapy, the effect of systemic therapy, or spontaneous loss [7]. Our findings indicate that the loss of ER is not associated with endocrine therapy, which does not support the hypothesis that the selective pressure of endocrine therapy causes loss. Additionally, we were unable to demonstrate an association with chemotherapy.

*ESR1* mutation was found in 13% of the metastases, which is similar to the 12.1% reported by Niu and somewhat lower than the 21% reported by Jeselsohn [18,19]. Others reported a frequency of 16% in distant metastases, predominantly the p.(D538G) mutation [20]. In a large study, Toy et al. found *ESR1* mutations in 10% of their cases, with p.(D538G) being the most frequent [21]. However, they also included Her2-positive and triple-negative tumors. In the group of ER-positive Her2-negative tumors, the *ESR1* mutation rate was 13.5% [21]. Razavi et al. reported a frequency of 15%, including only patients with ER-positive tumors [8]. In the AMEERA-3 trial of a selective estrogen-receptor degrader in patients with metastatic ER-positive Her2 negative tumors, the *ESR1* mutation rate was 19.3% in the intervention arm and 18.4% in the control arm [14]. In the BOLERO-2 trial of the added value of everolimus in metastatic ER-positive Her2-negative tumors, the *ESR1* mutation rate was 28.8% in circulating cell-free DNA samples [22]. A recent review by Herzog et al. reported *ESR1* mutations in 11 to 54% of the metastases [23]. Variations among studies may be attributed to patient selection and sequencing methodology. Some methods are limited to detecting hotspot mutations, while others can identify additional mutations. We utilized the targeted sequencing of hotspot mutations of the helix 12 of the *ESR1* gene in formalin-fixed paraffin-embedded tumor samples. Disease stage is another factor to consider, since Zundelevich et al. demonstrated an *ESR1* mutation rate of 12% in newly diagnosed metastases and a rate of 18% in advanced metastatic breast cancer [20]. Others reported similar findings, including 10.7% and 19.9% [24]. The mutation rate also can be dependent upon the location of the biopsy. Liver metastases were the distant metastases that were most often mutated in two studies, reporting 21.3% and 30%. However, we did not find a difference between liver and non-liver metastases [21,25]. 

It is noteworthy that studies utilizing circulating DNA have reported the highest mutation rates [23]. In patients with advanced breast cancer who have undergone one or two lines of endocrine therapy, Briard et al. recently reported a 47.8% incidence of *ESR1* mutations based on circulating DNA analysis [15]. A pooled meta-analysis of liquid biopsies found a 23% mutation rate [26]. In a study of matched biopsies of metastases and circulating cell-free DNA, *ESR1* mutation was found in six biopsy samples and four corresponding blood samples. Additional *ESR1* mutations were found in three blood samples [27]. Others have reported a concordance of 47% between circulating cell-free DNA and metastases and also found a higher mutation rate in circulating cell-free DNA than in tissue [28]. Interestingly, the mutation rate in circulating tumor cells was found to be lower than in cell-free DNA [29].

We found that *ESR1* mutations can be found in 2 out of 123 (1.6%) of the primary tumors. A similarly low frequency of approximately 1% was found by Razavi et al., while the frequency in metastases was 15%, which is about the same as we observed [8]. Still, others have reported frequencies of between 0% and 3.5% [9,21,28,30].

We found that *ESR1* mutations only occur in ER-positive metastases. Several other authors reported that the ER expression of the metastases was positive [20,31]. A monoclonal antibody (SP1, Ventana) was used to detect ER expression. Therefore, it can be inferred that the binding site of this antibody is not affected by the conformational changes of the receptor protein induced by the *ESR1* mutation. This finding seems to suggest that *ESR1* mutation and ER loss are mutually exclusive, similar to the study of *ESR1* mutations by Razavi et al. that demonstrated this for several mutations involved in the mitogen-activated protein kinase (MAPK) pathway, *ERBB2* mutations, *NF-1* loss-of-function mutations, and alterations in genes involved in the regulation of estrogen-receptor transcription [8].

A statistically significant correlation was observed between the loss of ER in metastatic lesions and overall survival following biopsy. This has been observed in several other studies [3,32,33,34]. A recent meta-analysis reported a hazard ratio of 1.67 (confidence interval 1.67–2.04), which is somewhat lower than the hazard ratio of 3.3 (confidence interval 1.9–5.5) that we found [35]. The analysis did not reveal a statistically significant association between *ESR1* mutation status and survival. In a retrospective analysis of the BOLERO-2 study, patients with an *ESR1* mutation exhibited a shorter survival period [22]. A retrospective analysis of the SOFEA and EFFECT trial demonstrated shorter survival of patients with an *ESR1* mutation undergoing exemestane treatment but not for patients with an *ESR1* mutation undergoing fulvestrant treatment [36]. In the AMEERA 3 trial, no effect was observed in patients for a selective estrogen-receptor degrader, although the data were immature [14]. A recent meta-analysis reported a hazard ratio of 1.59 [37]. The discrepancies observed in these studies can be attributed to differences in systemic therapy. It is also notable that the majority of these studies employed the measurement of *ESR1* mutations in cell-free plasma DNA. The study by Zundelevich employed DNA extracted from tissue biopsies and reported an effect on progression-free survival [20]. The overall survival was not reported. 

Time to progression has not been reported widely in the literature. For ER loss, we found a small difference in the Kaplan–Meier curves. However, this difference was not found in univariate and multivariate analyses. *ESR1* mutation did not show any difference in these analyses, but older age had a significant effect on TTP. Zundelevich et al. reported a shorter distant recurrence-free survival, as calculated from the time of diagnosis in patients with *ESR1* mutations [20].

One limitation of our study is that we used targeted sequencing of hotspot mutations, which may have caused us to miss some clinically relevant mutations, such as E380, S463, V422, G442, F461, and L469 [38]. Additionally, we only had access to one metastasis per patient, and several authors have reported that there can be discrepancies in ER loss between the metastases from different sites in the same patient [2,6]. *ESR1* mutations have been detected in circulating DNA, while there was no mutation detected in the available tumor biopsy, indicating the presence of nonbiopsied tumor sites with mutations [39]. This suggests that there may be differences in the *ESR1* mutations between different metastases, as demonstrated in an autopsy study of metastatic breast cancer [40]. This may lead to an underestimation of the mutation rate and may explain why we did not find an effect of *ESR1* mutation on overall survival. Two of the *ESR1* mutations that we detected, p.(D538G) and p.(Y537S), are known for their constitutive activity, whereas p.(L536H) only leads to estrogen hypersensitivity [21]. Since we found only two p.(L536H) mutations out of 18 *ESR1* mutations, we were unable to test if these mutations would make a difference in TTP or OS. As this was a retrospective study, we did not have enough imaging data to determine progression-free survival.

## 5. Conclusions

In conclusion, we have demonstrated that ER loss and *ESR1* mutation can explain resistance to endocrine therapy in 30% of patients. ER loss has a negative effect on overall survival. Additionally, we found that ER loss and *ESR1* mutation are mutually exclusive.

## Figures and Tables

**Figure 1 cancers-16-03025-f001:**
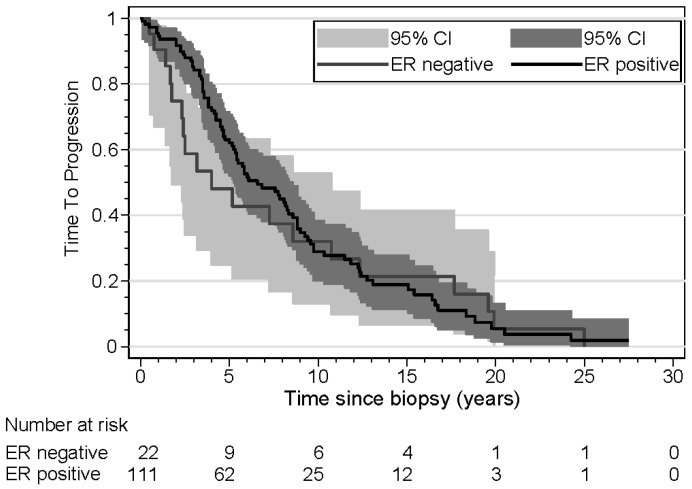
Kaplan–Meier curves displaying time to progression of patients (n = 133) with ER-positive and ER-negative metastases.

**Figure 2 cancers-16-03025-f002:**
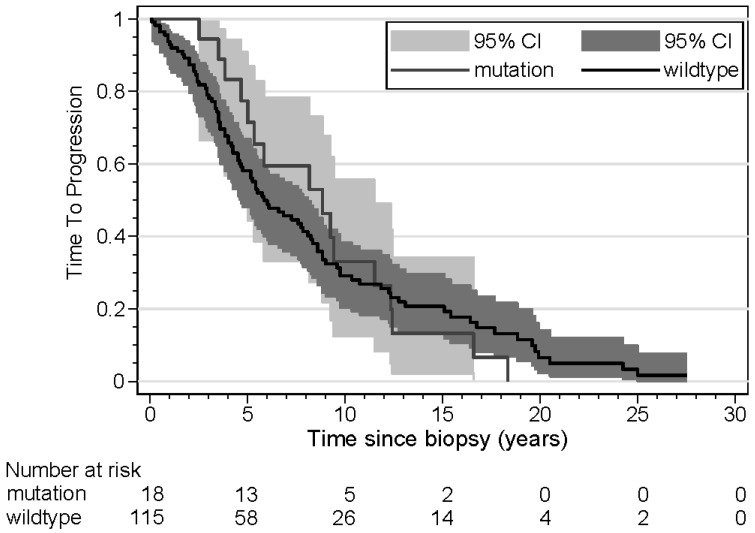
Kaplan–Meier curves displaying time to progression of patients (n = 133) with *ESR1* wildtype and *ESR1* mutated metastases.

**Figure 3 cancers-16-03025-f003:**
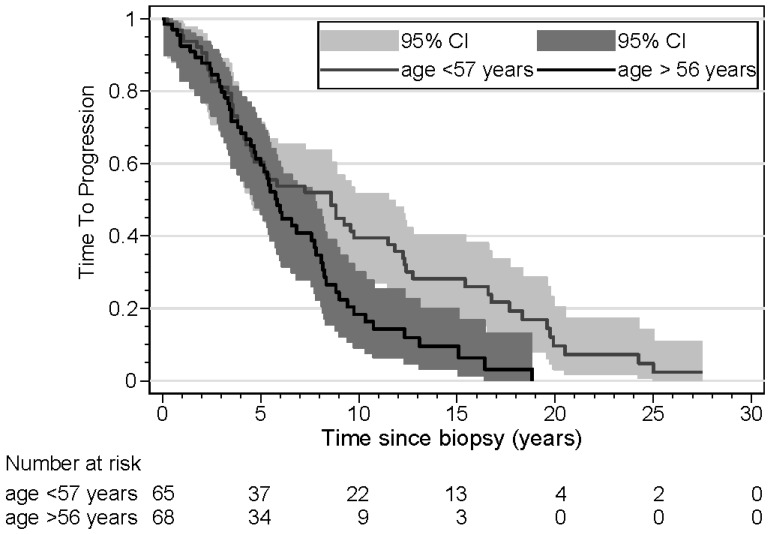
Kaplan–Meier curves displaying time to progression of patients (n = 133) aged ≤56 years and ≥57 years.

**Figure 4 cancers-16-03025-f004:**
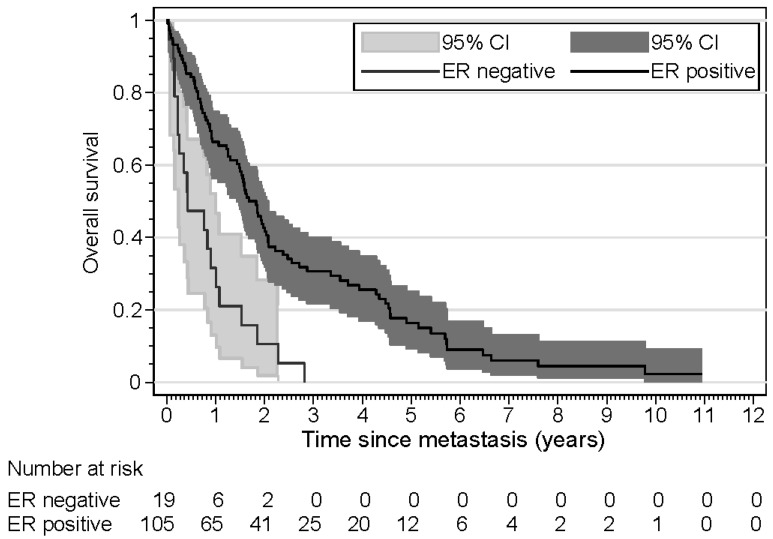
Kaplan–Meier curves displaying the overall survival of patients (n = 123) with ER-positive and ER-negative metastases.

**Figure 5 cancers-16-03025-f005:**
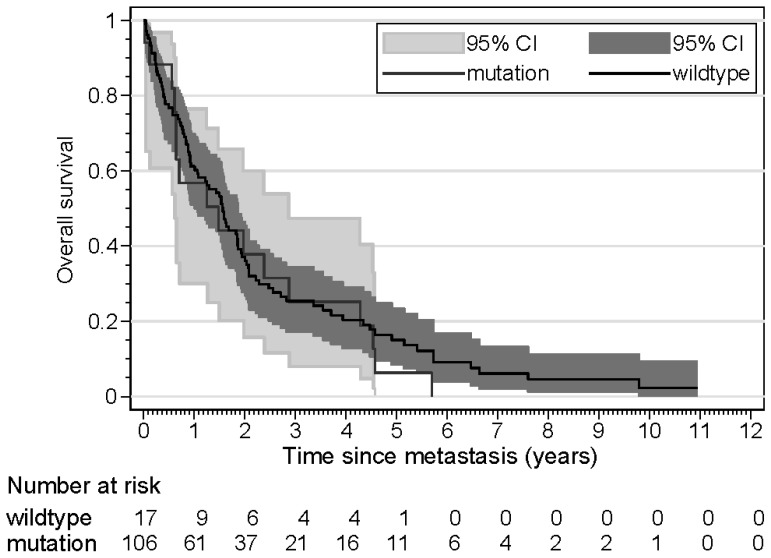
Kaplan–Meier curves displaying the overall survival of patients (n = 123) with *ESR1* wildtype and *ESR1* mutated metastases.

**Figure 6 cancers-16-03025-f006:**
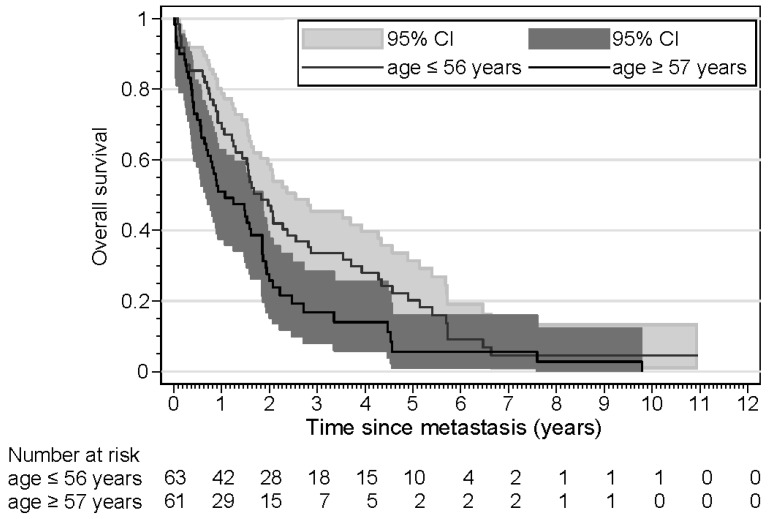
Kaplan–Meier curves displaying the overall survival of patients (n = 123) aged ≤ 56 years and ≥57 years.

**Table 1 cancers-16-03025-t001:** Patient characteristics.

		N	%
Median Age		57 (33–83)	
T status	Tis	0	0
	T1	56	41
	T2	40	29
	T3	12	9
	TX	28	21
N status	N0	40	29
	N1	33	24
	N2	20	15
	N3	12	9
	NX	31	23
Histology	NST	96	70
	ILC	28	20
	Other	4	3
	Unknown	8	6
Grade	1	8	6
	2	67	49
	3	28	21
	Unknown	33	24
Endocrine therapy	yes	89	65
	no	34	25
	Unknown	13	10
Chemotherapy	Yes	64	47
	No	61	45
	Unknown	11	8
Site of metastasis	liver	71	52
	lung	15	11
	skin	19	14
	other	31	23

T and N status according to the TNM classification edition 8; NST = no special type; ILC = infiltrating lobular cancer.

**Table 2 cancers-16-03025-t002:** The effect of endocrine therapy on *ESR1* mutation.

	Endocrine Therapy			
*ESR1*	No	Yes	Unknown	Total
Wildtype	34	71	13	118
Mutation	0	18	0	18
total	34	89	13	136

**Table 3 cancers-16-03025-t003:** The effect of endocrine therapy on ER loss.

	Endocrine Therapy			
ER loss	No	Yes	Unknown	Total
Yes	6	12	5	23
No	28	77	8	113
total	34	89	13	136

**Table 4 cancers-16-03025-t004:** Time to progression.

Time ^^^ to Metastasis, Percentiles		25%	50%	75%
Mutation metastasis	wildtype	3.4	5.8	12.3
	mutation	5	8.8	12.3
ER metastasis	positive	3.8	6.6	12.3
	negative	1.8	4	12.3
Age at biopsy ^#^	≤56	3.6	8.6	16.6
	≥57	3.4	5.8	8.8
T size at excision ^#^	≤20	5.2	9.3	15.4
	≥21	3.3	5.2	8.6
**Univariate Analysis**		**RR ***	**95% CI**	***p*-Value**
Mutation metastasis	wildtype	1		
	mutation	0.95	0.56–1.61	0.84
ER metastasis	positive	1		
	negative	1.15	0.70–1.89	0.57
Age at biopsy	continuous	1.02	1.002–1.04	0.03
Age at biopsy ^#^	≤56	1		
	≥57	1.43	0.98–2.08	0.07
T size at excision	continuous	1.02	1.003–1.03	0.02
T size at excision ^#^	≤20	1		
	≥21	1.86	1.21–2.86	0.01
**Multivariate Analysis ^+^**		**HR ^&^**	**95% CI**	***p*-Value**
ER metastasis	positive	1		
	negative	1.89	1.11–3.23	0.02
Age at biopsy	continuous	1.05	1.03–1.07	<0.001
T size at excision	continuous	1.02	1.01–1.03	<0.001

^: in years; #: grouped at median; *: Mantel–Haenszel hazard rate ratio; ^&^: Cox regression hazard ratio; +: final model.

**Table 5 cancers-16-03025-t005:** Overall survival.

Time ^^^ to Death after Metastasis, Percentiles		25%	50%	75%
Mutation metastasis	wildtype	0.6	1.6	3.4
	mutation	0.6	1.5	4.3
ER metastasis	positive	0.7	1.7	4.3
	negative	0.2	0.4	1.1
Age at biopsy ^#^	≤56	0.9	1.8	4.3
	≥57	0.4	1.1	2.1
**Univariate Analysis**		**RR ***	**95% CI**	***p*-Value**
Mutation metastasis	wildtype	1		
	mutation	1.15	0.68–1.95	0.61
ER metastasis	positive	1		
	negative	3.21	1.95–5.26	<0.001
Age at biopsy	continuous	1.02	0.998–1.03	0.08
Age at biopsy ^#^	≤56	1		
	≥57	1.66	1.14–2.42	0.01
**Multivariate Analysis ^+^**		**HR ^&^**	**95% CI**	***p*-Value**
ER metastasis	positive	1		
	negative	1.89	1.11–3.23	0.02
Age at biopsy	continuous	1.05	1.03–1.07	<0.001

^: in years; #: grouped at median; *: Mantel–Haenszel hazard rate ratio; ^&^: Cox regression hazard ratio; +: final model.

## Data Availability

The data presented in this study are available on request from the corresponding author due to legal restrictions.

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
