# Peer review of "Estrogen-Receptor Loss and ESR1 Mutation in Estrogen-Receptor-Positive Metastatic Breast Cancer and the Effect on Overall Survival"

_cancers, 2024, doi:10.3390/cancers16173025_

Round 1
Reviewer 1 Report (Previous Reviewer 1)
Comments and Suggestions for Authors
I thank the authors for their work in answering Reviewers' comments and questions.
I still have some minor points to be addressed:
- Abstract. Please add a conclusion statement.
- M&M section. Please describe how you did the ESR1 evaluation, providing the experimental details (including the procedure for the staining, antibody name (manufacturer, clone), reagents, and type of microscope used for evaluation, if different pathologists reviewed each case, etc.).
- Figures 4-5. Please change the white color of the CI. It is not evident compared to the background.
Author Response
Please see the attachment

Reviewer 2 Report (Previous Reviewer 2)
Comments and Suggestions for Authors
The authors have addressed most of my concerns. The reviewer finds the revised version of the manuscript to be improved in term of data presentation and organization and approves it for the publication in the Cancers journal after fixing still remaining mistakes:
Comments on the Quality of English Language
The reviewer finds the English language quality to be suitable. Just minor editing of style is required.
Author Response
Please see the attachment

This manuscript is a resubmission of an earlier submission. The following is a list of the peer review reports and author responses from that submission.
Round 1
Reviewer 1 Report
Comments and Suggestions for Authors
The work of Westenend and colleagues aims to investigate the potential association between ER loss and ESR1 mutation in ER+Her2- metastatic breast cancer patients.
The main authors’ findings are that: i) ER loss and ESR1 are mutually exclusive genetic alterations (several other studies already reported this and the authors themselves correctly cite it); and ii) ER loss, but not ESR1 mutation, correlated with a different patient overall survival.
Because there are a huge number of studies investigating ESR1/ER in this setting, the work's novelty is limited overall.
Major comments
- The authors should add progression-free survival (PFS) analysis. This would increase the impact of their study.
- There is no shown correlation between the types of ESR1 mutation and patient clinical outcome. It is already known that different mutations have a distinct impact on endocrine therapy resistance (see for example, doi: 10.1158/2159-8290.CD-15-1523): the most common ESR1 mutations, such as D538G and Y537, are known for their constitutive activity, whereas other variants only lead to estrogen hypersensitivity. The authors must include in their analysis if distinct variants are associated with a different patient PFS, endocrine therapy resistance, and OS.
- The authors identified age as a significant factor in OS analysis. I suggest they show these results in additional figures/tables.
Minor comments
- Sentences in the abstract are repeated (for example that ESR1 and ER loss are mutually exclusive). Please remove them.
- Please add in the Material and Methods section ER loss evaluation in FFPE samples.
- Line 148. “The overall survival was 61% at 1 year, 25% at 3 years, 14% at 5 years”. It is unclear if this is referring to the entire cohort or just to a subgroup of patients in Figure 1 or 2. Please specify it.
- Please add age information in Table 1 (Clinical characteristics of the population)
Comments on the Quality of English Language
Minor editing of English language required
Author Response
Response to reviewer 1 c 1. Summary |
|
|
|||||||||||||||||||||||||||||||||||||||||||||||||||||||||||||||||||||||||||||||||||||||||||||||||||||||||||||||||||||||||||||
Thank you very much for taking the time to review this manuscript. Please find the detailed responses below and the corresponding revisions/corrections highlighted/in track changes in the re-submitted files.
|
|||||||||||||||||||||||||||||||||||||||||||||||||||||||||||||||||||||||||||||||||||||||||||||||||||||||||||||||||||||||||||||||
2. Questions for General Evaluation |
Reviewer’s Evaluation |
Response and Revisions |
|||||||||||||||||||||||||||||||||||||||||||||||||||||||||||||||||||||||||||||||||||||||||||||||||||||||||||||||||||||||||||||
|
|
|
|||||||||||||||||||||||||||||||||||||||||||||||||||||||||||||||||||||||||||||||||||||||||||||||||||||||||||||||||||||||||||||
|
|
|
|||||||||||||||||||||||||||||||||||||||||||||||||||||||||||||||||||||||||||||||||||||||||||||||||||||||||||||||||||||||||||||
Is the research design appropriate? |
Must be improved |
We added further data on TTP as a response to the request to add PFS |
|||||||||||||||||||||||||||||||||||||||||||||||||||||||||||||||||||||||||||||||||||||||||||||||||||||||||||||||||||||||||||||
Are the methods adequately described? |
Must be improved |
We clarified that ER loss was evaluated in FFPE samples. The method section has been expanded with the TTP analysis |
|||||||||||||||||||||||||||||||||||||||||||||||||||||||||||||||||||||||||||||||||||||||||||||||||||||||||||||||||||||||||||||
Are the results clearly presented? |
Must be improved |
Table 1 was revised and additional tables were added. We also added more graphs |
|||||||||||||||||||||||||||||||||||||||||||||||||||||||||||||||||||||||||||||||||||||||||||||||||||||||||||||||||||||||||||||
Are the conclusions supported by the results? |
Can be improved |
More details of the data analysis are added in Table 5 and figures 4-6. |
|||||||||||||||||||||||||||||||||||||||||||||||||||||||||||||||||||||||||||||||||||||||||||||||||||||||||||||||||||||||||||||
3. Point-by-point response to Comments and Suggestions for Authors |
|||||||||||||||||||||||||||||||||||||||||||||||||||||||||||||||||||||||||||||||||||||||||||||||||||||||||||||||||||||||||||||||
Comments 1: The authors should add progression-free survival (PFS) analysis. This would increase the impact of their study. |
|||||||||||||||||||||||||||||||||||||||||||||||||||||||||||||||||||||||||||||||||||||||||||||||||||||||||||||||||||||||||||||||
Response 1: Thank you for pointing this out. We reported overall survival since the time of the metastasis biopsy and considered adding progression-free survival. In order to do so we would need imaging data to document how long metastases were stable or in regression before there was disease progression. Since this is a retrospective study, imaging data for many patients are missing. We added a line in the discussion (348-349): “As this was a retrospective study, we did not have enough imaging data to determine progression-free survival.”
However, we do have the data to determine the time to progression (TTP). Therefore, we have added this analysis in the manuscript as follows: · Introduction, last parapgraph line 81: time to progression added as an aim of the study · Method, statistics paragraph, line 127-128: “Time to progression was calculated from the date of biopsy of the primary tumor to the date of the metastasis biopsy,” · Results, third paragraph, line 174-182: “Results of the time to progression (TTP) analyses are shown in Table 4. Three patients were excluded from this analysis because they presented with synchronous metastasis. Age was a significant risk factor with older patients demonstrating a shorter time to progression both on univariate and multivariate analysis. Tumor size was another significant risk factor, patients with a larger tumor had a shorter TTP on univariate and multivariate analysis. In patients with ER loss of the metastasis, TTP was not significantly shorter on univariate analysis, but in a multivariate analysis taking into account age and tumor size, TTP was significantly shorter (HR 1.89). Kaplan-Meier curves for TTP according to ESR1 mutation, ER loss and age are shown in Figures 1-3.” · Results line 208-209, Table 4 Table 4. Time To Progression
^ in years, # grouped at median, * Mantel-Haenzel harzard rate ratio, & Cox regression hazard ratio, + final model Results, lines 214-223 TTP figures 1, 2 and 3Figure 1. Kaplan-Meier curves displaying time to progression of patients (n = 133) with ER positive and ER negative metastases Figure 2. Kaplan-Meier curves displaying time to progression of patients (n = 133) with ESR1 wild type and ESR1 mutated metastases. · Figure 3. Kaplan-Meier curves displaying time to progression of patients (n=133) aged ≤ 56 years and ≥ 57 years.
· Discussion, paragraph 8, lines 328-333: “Time to progression has not been reported widely in the literature. For ER-loss we found a small difference in the Kaplan-Meier curves, however this difference was not found on univariate and multivariate analysis. ESR1 mutation did not show any difference in these analyses, but older age had a significant effect on TTP. Zundelevich et al. reported a shorter distant recurrence free survival as calculated from the time of diagnosis in patients with ESR1 mutations [19].”
|
|||||||||||||||||||||||||||||||||||||||||||||||||||||||||||||||||||||||||||||||||||||||||||||||||||||||||||||||||||||||||||||||
Comments 2: There is no shown correlation between the types of ESR1 mutation and patient clinical outcome. It is already known that different mutations have a distinct impact on endocrine therapy resistance (see for example, doi: 10.1158/2159-8290.CD-15-1523): the most common ESR1 mutations, such as D538G and Y537, are known for their constitutive activity, whereas other variants only lead to estrogen hypersensitivity. The authors must include in their analysis if distinct variants are associated with a different patient PFS, endocrine therapy resistance, and OS.
|
|||||||||||||||||||||||||||||||||||||||||||||||||||||||||||||||||||||||||||||||||||||||||||||||||||||||||||||||||||||||||||||||
Response 2: We do agree that it would be interesting to explore if there is a difference in PFS, endocrine therapy resistance and OS according to the specific mutation that we found. However, we found 9 D538G, 7 Y537 and 2 L536H mutations which means 16 mutations with constitutive activity and 2 with hypersensitivity. Therefore, the study is underpowered to test for differences in these types of mutations. We have addressed this point in the discussion of the limitations of the study. Discussion, paragraph 9, lines 344-349: “Two of the ESR1 mutations that we detected, p.(D538G) and p.(Y537S) are known for their constitutive activity, whereas p.(L536H) only leads to estrogen hypersensitivity [20]. Since we found only 2 p.(L536H) mutations out of 18 ESR1 mutations, we were unable to test if these mutations would make a difference in TTP or OS. Since this was a retrospective study, we did not have enough imaging data to determine progression free survival.”
Comments 3: The authors identified age as a significant factor in OS analysis. I suggest they show these results in additional figures/tables. Response 3: We agree that the presentation of these results can be improved. We therefore added Table 4 and Figure 6.
· Table 5. Overall survival
^ in years, # grouped at median, * Mantel-Haenzel harzard rate ratio, & Cox regression hazard ratio, + final model Figure 6. Kaplan-Meier curves displaying overall survival of patients (n=123) aged ≤ 56 years and ≥ 57 years.
|
|||||||||||||||||||||||||||||||||||||||||||||||||||||||||||||||||||||||||||||||||||||||||||||||||||||||||||||||||||||||||||||||
Comment 4: Sentences in the abstract are repeated (for example that ESR1 and ER loss are mutually exclusive). Please remove them. Response 4: We agree that there were some redundant sentences. We have removed the following: · Line 15 “Several mechanisms for resistance to endocrine therapy have been described” · Line 16 “Two of these mechanisms are” · Line 23 “In 41 metastases (30%) we were able to demonstrate a known mechanism of resistance to endocrine therapy” · Line24 “ESR1 mutation and ER loss are mutually exclusive” Comment 5: Please add in the Material and Methods section ER loss evaluation in FFPE samples. Response 5: We agree and added the following to Materials an Methods, paragraph 1, line 90-91: “Estrogen receptor expression was determined in slides from formalin-fixed, paraffin-embedded tissue.” Comment 6: - Line 148. “The overall survival was 61% at 1 year, 25% at 3 years, 14% at 5 years”. It is unclear if this is referring to the entire cohort or just to a subgroup of patients in Figure 1 or 2. Please specify it. Response 6: This is indeed unclear. We have deleted this sentence. More detailed information is given in the upper box of table 5.
^ in years, # grouped at median, * Mantel-Haenzel harzard rate ratio, & Cox regression hazard ratio, + final model Comment 7: Please add age information in Table 1 (Clinical characteristics of the population) Response 7: We added age in the first line of table 1.
Response to Comments on the Quality of English Language |
|||||||||||||||||||||||||||||||||||||||||||||||||||||||||||||||||||||||||||||||||||||||||||||||||||||||||||||||||||||||||||||||
Point 1: Minor editing of English language required
|
|||||||||||||||||||||||||||||||||||||||||||||||||||||||||||||||||||||||||||||||||||||||||||||||||||||||||||||||||||||||||||||||
Response 1: As suggested by reviewer 2 we have used synonyms for the frequently used word study and sometimes slightly rephrased sentences to avoid the use of study.
|
|||||||||||||||||||||||||||||||||||||||||||||||||||||||||||||||||||||||||||||||||||||||||||||||||||||||||||||||||||||||||||||||
Response 1: As suggested by reviewer 2 we have used synonyms for the frequently used word study and sometimes slightly rephrased sentences to avoid the use of study.
|
Reviewer 2 Report
Comments and Suggestions for Authors
The manuscript at hand concerning the role of ER loss and ESR1 point mutations in human metastatic breast malignancies is well written and clear. However, there are important issues that speak against its publication in the Cancers journal in present form.
The reviewed work observes mutations limited to helix 12 of the ESR1 gene, while the literature describes a number of this gene mutations located in other sites (f.ex. E380, S463, V422, G442, F461, L469) and also playing a significant role in the breast tumor progression. Were the point mutations listed above detected by sequencing performed in the present investigation?
The Abstract exceeds the 200-word limit established in the journal's “Instructions for Authors” (222 words).
References do not meet the the journal's “Instructions for Authors”.
The authors must rewrite the manuscript using synonyms of the term 'study' which was repeated about 40 times within the text.
Comments on the Quality of English Language
The reviewer finds the English language quality to be almost suitable. Just moderate editing of style is required.
Author Response
Response to Reviewer 2 Comments
|
||
1. Summary |
|
|
Thank you very much for taking the time to review this manuscript. Please find the detailed responses below and the corresponding revisions/corrections highlighted/in track changes in the re-submitted files |
||
2. Questions for General Evaluation |
Reviewer’s Evaluation |
Response and Revisions |
Does the introduction provide sufficient background and include all relevant references? |
Can be improved |
We made some minor modifications |
Are all the cited references relevant to the research? |
Can be improved |
We found that reference 8 and 21 are the same, this has been corrected |
Is the research design appropriate? |
Must be improved |
We added further data on TTP as a response to the request of reviewer 1 to add PFS |
Are the methods adequately described? |
Can be improved |
We clarified that ER loss was evaluated in FFPE samples. The method section has been expanded with the TTP analysis in response to a suggestion by reviewer 1 |
Are the results clearly presented? |
Can be improved |
Table 1 was revised and additional tables were added. We also added more graphs |
Are the conclusions supported by the results? |
Must be improved |
More details of the data analysis are added in Table 5 and figures 4-6. |
3. Point-by-point response to Comments and Suggestions for Authors |
||
Comments 1: The reviewed work observes mutations limited to helix 12 of the ESR1 gene, while the literature describes a number of this gene mutations located in other sites (f.ex. E380, S463, V422, G442, F461, L469) and also playing a significant role in the breast tumor progression. Were the point mutations listed above detected by sequencing performed in the present investigation?
|
||
Response 1: Thank you for pointing this out. As already mentioned in Material and Methods, 2.3 Amplification and Pyrosequensing, line 113 our mutation analysis was limited to codons 536-538. Although this is a hotspot for mutations in the ESR1 gene, we may have missed some mutations like the ones mentioned by the reviewer. As this is a limitation of our study, be added the following to the Discussion, line 334-336: “One limitation of our study is that we used targeted sequencing of hotspot mutations, which may have caused us to miss some clinically relevant mutations like E380, S463, V422, G442, F461, L469.” |
||
Comments 2: The Abstract exceeds the 200-word limit established in the journal's “Instructions for Authors” (222 words).
|
||
Response 2: Agree. We have reduced the number of words by removing redundant sentences. · Line 15 “Several mechanisms for resistance to endocrine therapy have been described” · Line 16 “Two of these mechanisms are” · Line 23 “In 41 metastases (30%) we were able to demonstrate a known mechanism of resistance to endocrine therapy” · Line24 “ESR1 mutation and ER loss are mutually exclusive” Comment 3: References do not meet the journal's “Instructions for Authors”. Response 3: We have changes the format of the references according to the journal’s instructions Comment 4: The authors must rewrite the manuscript using synonyms of the term 'study' which was repeated about 40 times within the text. Response 4: Agree, we have used synonyms for the frequently used word study and sometimes slightly rephrased sentences to avoid the use of study.
|
||
4. Response to Comments on the Quality of English Language |
||
Point 1: The reviewer finds the English language quality to be almost suitable. Just moderate editing of style is required.
|
||
Response 1: As mentioned above, we have made some slight modifications to improve the style. |
||
5. Additional clarifications In preparing the references according to the journal’s instruction we noticed that reference 8 and 21 are the same. The numbering of the refences in the text have been corrected for this mistake. |
||
|
